# A BIRD'S EYE VIEW ON COHERENCE, AND A WORM'S EYE VIEW ON COHESION

## ABSTRACT

Generating coherent and cohesive long-form texts is a challenging problem in natural language generation. Previous works relied on a large amount of human-generated texts to train neural language models, however, few attempted to explicitly model the desired linguistic properties of natural language text, such as coherence and cohesion using neural networks. In this work, we train two expert discriminators for coherence and cohesion to provide hierarchical feedback for text generation. We also propose a simple variant of policy gradient, called *negative-critical sequence training* in which the reward *baseline* is constructed from randomly generated negative samples. We demonstrate the effectiveness of our approach through empirical studies, showing improvements over the strong baseline – attention-based bidirectional MLE-trained neural language model – in a number of automated metrics. The proposed model can serve as baseline architectures to promote further research in modeling additional linguistic properties for downstream NLP tasks.

## 1 INTRODUCTION

The terms *coherence* and *cohesion* in linguistics are commonly defined as follows (Williams & Colomb, 1995).

> *Cohesion*: sentence pairs fitting together the way two pieces of a jigsaw puzzle do.
> *Coherence*: what all the sentences in a piece of writing add up to, the way all the pieces in a puzzle add up to the picture on the box.

In layman's terms, *cohesion* indicates that two consecutive sentences are *locally* well-connected, and *coherence* indicates that multiple sentences *globally* hold together.

Generating cohesive and coherent natural language texts that span multiple sentences is a challenging task mainly due to two reasons. First, there is no formal specification of cross-sentence linguistic properties, such as coherence and cohesion of a text. Second, there is no widely accepted surrogate neural model to evaluate the two properties.

Most state-of-the-art neural approaches to natural language generation (NLG) relied on a large amount of human-generated texts to train language models (Cho et al., 2014; Graves, 2013; Sutskever et al., 2014). Although these models can generate sentences that, if judged individually, are similar to human-generated ones, they often fail to capture the local and global dependencies among sentences, resulting in a neither coherent nor cohesive text. For example, neural language models based on Recurrent Neural Networks (RNNs) are widely applied to response generation for dialogue (Vinyals & Le, 2015; Shang et al., 2015; Sordoni et al., 2015; Li et al., 2015). Although the responses by themselves look reasonable, they are either bland such as "I don't know", or incoherent with the whole dialogue session. See Gao et al. (2018) for a comprehensive survey.

In this paper, we strive to address the challenge in a principled manner. We propose a pair of discriminators to score whether and to what extent a text is coherent or cohesive. The coherence discriminator measures the compatibility among all sentences in a text using sentence-level features, thus providing a macro-level view on the multi-sentence text. The cohesion discriminator, on the other hand, measures the compatibility of each pair of consecutive sentences using only word-level features, thus providing a micro-level view on neighboring sentences. These models, given a conditional input text and multiple candidate output texts, are learned to score the candidates with respect

to the criterion by optimizing pairwise ranking losses. These scores are then used as reward signals to train an RNN-based language model to generate (more) coherent and cohesive texts.

Our main contributions are three-fold: (1) we propose two linguistic discriminators for modeling coherence and cohesion of a text; (2) we present a simple yet effective training mechanism to encode these linguistic properties; and (3) we propose *negative-critical sequence training*, a variant of policy gradient method, which uses negative samples to construct its reward *baseline*.

This paper proposes a neural approach to explicitly model cross-sentence linguistic properties, i.e., coherence and cohesion, for long text generation. Despite the encouraging initial results, we only scratched the surface of the problem. The proposed method is yet to be significantly improved to meet the ultimate goal of generating meaningful and logical long-form texts. We cast the text generation as an RL problem and review recent work in Section 2, and detail our approach in Section 3.

## 2 RELATED WORK

**Coherence and cohesion.** Coherence and cohesion have been extensively studied in the computational linguistics community, particularly in the 'pre-deep-learning' era. Lack of formal specifications for coherence and cohesion (Mani et al., 1998), resulted in many different formalism, such as Rhetorical Structure Theory (Mann & Thompson, 1988), and other forms of coherence and cohesion relations and their quantification (Mani et al., 1998; Hobbs, 1985; Hovy, 1988; McKeown, 1985; Cohen & Levesque, 1985; Hovy, 1991; Cristea et al., 1998; Halliday & Hasan, 1996; Liddy, 1991; Van Dijk, 2013; Edmundson, 1969; Barzilay & Lapata, 2008). In fact, the list is not exhaustive. However, modeling coherence and cohesion of a text using models parametrized by neural networks have not been previously explored.

**Word sequence generation in a reinforcement learning framework.** A word sequence generation task can be framed as a reinforcement learning (RL) problem, in which the generator $G$ is acting as a *policy* $\pi$, with parameters $\theta_\pi$, and each generated word at time $t$, $w_t$, can be viewed as an action to be chosen by the policy from a large discrete space, or vocabulary, conditioned on state $s_{t-1} = w_{\leq t-1}$, which encodes the previously generated text sequence.

Let $r_t$ be the reward for a partially generated text sequence $w_{\leq t}$. We define the long-term expected reward $\mathcal{J}(\pi) = \mathbb{E}_{s_0 \sim q, \pi}[\sum_{t=1}^{\infty} \gamma^{t-1} r_t]$, where $q$ is the initial distribution of conditional input texts. Following Sutton et al. (1999), the gradient of $\mathcal{J}$ with respect to $\theta_\pi$ is

$$\nabla_{\theta_\pi} \mathcal{J} = \mathbb{E}_{s \sim \rho^\pi, a \sim \pi(\cdot|s)}[Q^\pi(s, a) \nabla_{\theta_\pi} \log \pi_{\theta_\pi}(a|s)]$$

where $\rho^\pi$ is the stationary distribution and $Q^\pi(s, a)$ is the expected return from state $s$ and taking action $a$, both following policy $\pi$. For brevity, we omit the derivation. In our work, we formulate text generation as an episodic RL problem with episode length $L$, rewards $r_L$ being available only at the end of episode and $\gamma = 1$.

There are many works on training neural language models using reward signals, such as Ranzato et al. (2015) and Paulus et al. (2017). These works directly optimize for specific metrics, such as BLEU (Papineni et al., 2002) or ROUGE (Lin & Hovy, 2003), using REINFORCE (Williams, 1992; Sutton et al., 1999). However, it is well-known that these metrics do not give a complete picture on the quality of the generated text. Only recently have there been efforts to provide more relevant quality objectives for which to optimize (Li et al., 2015; 2016a; Holtzman et al., 2018) the quality of interest such as consistency, repetition of text. But these works use the objective function to re-rank candidate outputs, not to reward or penalize outputs when they are generated in the first place. Li et al. (2016b) constructed a set of reward models, such as information flow and semantic coherence, to tune the generator, yet they do not provide an ablation study to elaborate the relative contribution of these reward models individually.

**GANs for text generation.** Another line of research is to use Generative Adversarial Networks (GANs) (Goodfellow et al., 2014) to incorporate feedback signals for text generation (Yu et al., 2017; Lin et al., 2017; Zhang et al., 2017c; Guo et al., 2017; Fedus et al., 2018; Zhang et al., 2018). However, the discriminator in these works are trained to distinguish real texts from the generated ones, operating as a black-box rather than providing fine-grained feedback on particular linguistic aspects of the texts. In fact, Yang et al. (2018) has partially addressed this issue by using a trained language model as the discriminator. Although the discriminator provides a fine-grained feedback at

the word-level, it does not critique on many important linguistic properties of generated texts, such as cohesion and coherence.

These text generators, when facing a long-form text generation task that span multiple sentences, are by no means perfect and often exhibit some critical errors, such as a breakdown of local connections between consecutive sentences (cohesion), let alone globally solid intention (coherence). As a result, readers can easily take these cues and discriminate such generated texts against real texts. In this paper, we argue that the primary reason is the lack of an effective mechanism of measuring and controlling the text quality in the generation process. The method we propose in the next section is intended to address the problem.

## 3 MODEL

We assume that global coherence of a text depends to a large degree upon how its individual sentences with different meanings are organized. So we focus our evaluation of coherence solely on the sentence-level. If the sentences are not organized properly, the intention of the paragraph as a whole is obscure, regardless of seamless local connectivity between consecutive sentences.

This is not to say that local connections between any two neighboring sentences should be overlooked. One can easily distinguish a model-generated sentence from a real one, simply by looking at whether the sentence followed by another sentence *relates*, besides grammar.

We instill these two different yet important concepts in two discriminators, each operating on the sentence level and word level. Our models closely resemble successful models for computer vision, such as StackGAN (Zhang et al., 2017a;b) and PatchGAN (Isola et al., 2017) in that they all provide hierarchical signals to their corresponding generators, where the signals are derived from raw low-level data. We call the sentence-level discriminator the *coherence* discriminator $D_{\text{coherence}}$, and the word-level discriminator the *cohesion* discriminator $D_{\text{cohesion}}$.

### 3.1 COHERENCE DISCRIMINATOR: $D_{\text{COHERENCE}}$

This discriminator measures how likely two text chunks form a coherent paragraph. Let $S := [s_1, s_2, ..., s_n]$ be the source text chunk that consists of $n$ sentences, $T := [t_1, t_2, ..., t_m]$ be the *real* target text chunk that consists of $m$ sentences, and $\widetilde{T} := [\widetilde{t}_1, \widetilde{t}_2, ..., \widetilde{t}_{\widetilde{m}}]$ be the *model-generated* target text chunk that consists of $\widetilde{m}$ sentences. $D_{\text{coherence}}$ is designed to distinguish a real pair $(S, T)$ from a synthetic pair $(S, \widetilde{T})$ by assigning them with different scores, i.e., $\text{Score}_{\text{coherence}}(S, T) > \text{Score}_{\text{coherence}}(S, \widetilde{T})$.

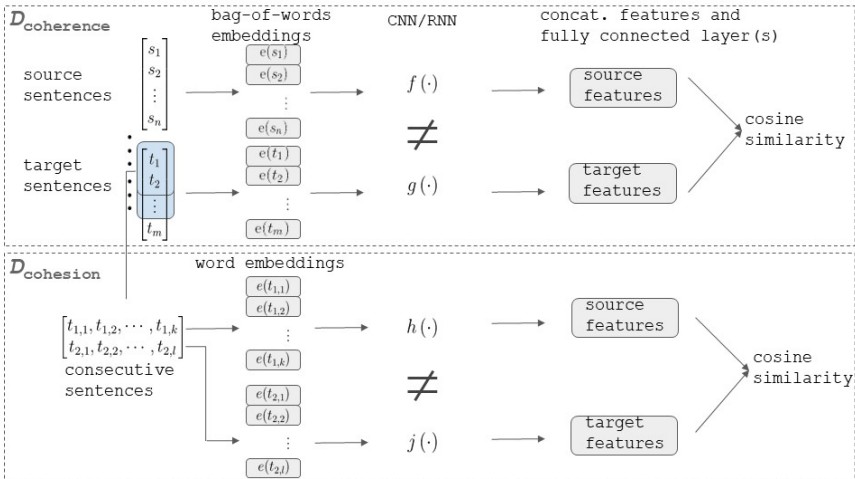

Figure 1: Illustration of coherence and cohesion discriminators. $D_{\text{coherence}}$ takes in bag-of-words sentence embeddings as inputs, and $D_{\text{cohesion}}$ takes in the raw word embeddings of consecutive sentences as inputs.

**Design**. Our design of $D_{\text{coherence}}$ is inspired by the Deep Structured Semantic Model (DSSM) (Huang et al., 2013; Gao et al., 2014; Xu et al., 2017). Given source text chunk $S$ and target text chunk $T$, $D_{\text{coherence}}$ computes their coherence score in two steps, as illustrated in Figure 1. First, a pair of convolutional networks (CNNs)[1] or RNNs are applied to encode both $S$ and $T$ into two low-dimensional continuous vectors. Second, the coherence score is computed as the cosine similarity of the two vectors. The score is a real value between $-1$ and 1, where 1 indicates the maximal coherence score, and $-1$ the minimal coherence score.

$D_{\text{coherence}}$ measures how likely $S$ and $T$ add up to a single coherent passage. The score depends on the parameters of the two CNNs/RNNs, or in other words, how $S$ and $T$ are encoded. Since we focus solely on the sentence-level features, we view a text chunk as a sequence of sentences, and view each sentence as a bag-of-words. Therefore, we represent each word using its pre-trained word embedding vector (Pennington et al., 2014) and represent each sentence using a vector which takes the average of its word embedding vectors. A text chunk is then represented as a sequence of sentence vectors which are fed to the CNN/RNN (either $f(\cdot)$ or $g(\cdot)$ in Figure 1). Notice that $f(\cdot)$ and $g(\cdot)$ do *not* share parameters. The parameters of $f(\cdot)$ and $g(\cdot)$ are optimized in such a way that a real pair scores higher than a synthetic pair:

$$\Delta(\theta_f, \theta_g) := D_{\text{coherence}}(f_{\theta_f}(S), g_{\theta_g}(T)) - D_{\text{coherence}}(f_{\theta_f}(S), g_{\theta_g}(\widetilde{T})) > 0$$

Formally, the task of optimizing $D_{\text{coherence}}$ can be stated as follows. Given a set of training samples of the form $((S, T), (S, \widetilde{T}))^{(i)}, i = 1...M$, we optimize parameters $(\theta_f, \theta_g)$ by minimizing the pairwise rank loss on training data defined as

$$\frac{1}{M} \sum_{i=1}^{M} L(\Delta(\theta_f, \theta_g)^{(i)})$$

where $L(\cdot)$ is a loss function, differentiable w.r.t. $(\theta_f, \theta_g)$.

In the following subsection, we will describe in turn how we construct the pairwise training samples and the form of the loss function. Since $D_{\text{coherence}}$ is used as a pairwise ranker, we employ the metrics commonly used in information retrieval for evaluation, such as recall at $K$ (R@$K$), which is defined as the fraction of correctly identifying an item in the TOP-$K$ retrieved list (Baeza-Yates & Ribeiro-Neto, 1999). We present the retrieval results on test data in Table 2.

**Training mechanism**. How do we construct these list of candidate target sentences $T$, given the source sentences $S$? We assume the $T$ that follows an $S$ in the data is a positive target sample, or the correct item to retrieve. Negative samples are constructed using three different methods within a batch while iterating through the training data, motivated by Wieting et al. (2016):

- Rotate $T$ with $S$ fixed (mismatch $S$ and $T$) across a batch. For a single $S$, this method yields $B - 1$ negative samples, where $B$ is the batch size.
- Shuffle the sentence order once, different from its original order, known as a derangement, in each $T$ to break coherence, and this yields one negative sample.
- Combine the previous two methods: rotate $T$ across a batch and shuffle sentences within $T$, yielding $B - 1$ negative samples.

These $2B - 1$ negative samples and a single positive sample, in sum, pose a significant challenge in learning. To fit this training task into a ranking framework, we optimize over

$$D_{\text{coherence}}(f_{\theta_f}(S), g_{\theta_g}(T)) - \text{AM}^{\lambda}_{i \in \{1, ..., 2B-1\}} \left[ D_{\text{coherence}}(f_{\theta_f}(S), g_{\theta_g}(\widetilde{T}_i)) \right]$$

where the arithmetic mean parametrized by $\lambda$: $\text{AM}^{\lambda}(x) = \sum_{i}^{N} w_i x_i$ and $w_i = e^{\lambda x_i} / \sum_{j} e^{\lambda x_j}$.

In our experiments, we fix $\lambda = 2$ and this assigns more weight to a more challenging negative sample[2]. Notice that $\text{AM}^{\lambda}$ is the *mean* if $\lambda = 0$, and approaches the *max* as $\lambda \to \infty$. Empirically, training the models using the weighted mean resulted in faster convergence, as opposed to using the single most challenging negative sample score (max) or the mean of all negative sample batch.

---

[1]We explored with deeper networks. However, the performance difference was marginal. For simplicity, we decided to use a 1-layer convolutional network architecture (Kim, 2014; Collobert et al., 2011).

[2]We performed a coarse grid search over the values of $\lambda$ and setting $\lambda = 2$ resulted in fast convergence to high recall scores on the dev dataset.

## 3.2 Cohesion discriminator: $D_{\text{cohesion}}$

Our second discriminator pays attention only to low-level features to capture a local connection between arbitrary two consecutive sentences because the pairwise connectivity influence readability.

For simplicity, $D_{\text{cohesion}}$ is similar to $D_{\text{coherence}}$, except that its architecture, input, and negative sample construction are modified to encode cohesion between any pair of sentences on the word-level. A single input sample to $D_{\text{cohesion}}$ is a pair of two consecutive sentences: $[s_{i,1}, s_{i,2}, ..., s_{i,n}]$ and $[s_{i+1,1}, s_{i+1,2}, ..., s_{i+1,m}]$, where $s_{i,k}$ denotes the $k$-th word in sentence $i$. We construct the negative samples using the three methods for training $D_{\text{coherence}}$, where shuffling occurs on the word level within each sentence, rather than shuffling multiple sentences on the sentence level.

## 3.3 Generator: $G$

The two pre-trained discriminators, $D_{\text{coherence}}$ and $D_{\text{cohesion}}$, are used to modify the text generation behavior of $G$. $G$ is an attention-based bidirectional sequence-to-sequence model. It is initially pretrained via maximizing the word-level likelihood given the training data, and we denote as $G_{\text{MLE}}$.

However, the model-generated texts from $G_{\text{MLE}}$ often do not hold to the standards of two discriminators. Therefore, we need to change the text generation behavior of $G$ with respect to the criteria. To this end, the scores from the discriminators that model the criteria are used as direct reward or penalty signals to modify the parameters of $G$. Given these signals, we use our proposed variant of the policy gradient, negative-critical sequence training, to apply parameter updates in $G$. We discuss the details in the next section.

## 4 Negative-Critical Sequence Training

Actor-critic methods (Barto et al., 1983; Witten, 1977) parameterized by neural networks typically require learning a separate critic network to estimate the expected future reward as a *baseline*, which in many cases is a difficult task by itself. In NLP, we have observed similar practices and challenges by Ranzato et al. (2015), Bahdanau et al. (2016), and Nguyen et al. (2017). However, recently Rennie et al. (2017) proposed an effective self-critical sequence training that avoids learning a separate critic network. Similarly, our method does not require learning a separate critic network, instead we directly use the scores of negative samples assigned by the discriminators as the *baseline*.

For an arbitrary pair of $S$ and $T_{gen}$, which is the generator's output conditioned on $S$, we compute the coherence and cohesion scores by calling $D_{\text{coherence}}$ and $D_{\text{cohesion}}$. Since each review consists of multiple sentences, the overall cohesion score is computed as the average of scores of all consecutive sentence pairs. These scalar scores, however, have no interpretation since the discriminators are trained by optimizing a margin ranking loss. Instead, the differences between positive sample scores and the maximal or average negative sample scores provide insight of how well the models can distinguish between the positives and the negatives. Therefore, these margins can be considered as rewards with baselines, and thus we define the reward functions as:

$$R_{\text{coherence}}(S, T) := D_{\text{coherence}}(f_{\theta_f}(S), g_{\theta_g}(T)) - \mathbb{E}_{\widetilde{T}}[D_{\text{coherence}}(f_{\theta_f}(S), g_{\theta_g}(\widetilde{T_i}))]$$
$$R_{\text{cohesion}}(s_i, s_{i+1}) := D_{\text{cohesion}}(h_{\theta_h}(s_i), j_{\theta_j}(s_{i+1})) - \mathbb{E}_{\widetilde{s}_{i+1}}[D_{\text{cohesion}}(h_{\theta_h}(s_i), j_{\theta_j}(\widetilde{s}_{i+1}))]$$

where $\tilde{\cdot}$ denotes a negative sample for a given source condition, and $\mathbb{E}_{\widetilde{T}}$ ( and $\mathbb{E}_{\widetilde{s}_{i+1}}$) are computed by averaging over an ensemble of negative samples. Notice that this reward resembles the ranking loss we use to train our discriminators, except that our baseline is an average score (instead of the weighted arithmetic mean) over negative samples. The rational for this difference is that: the maximal or the weighted arithmetic mean score baseline is in fact noisy to be used as rewards, because the best randomly constructed negative samples may be a *formidably* good sample. To alleviate such noise, we use the average discriminator scores of negative samples as the *baseline*, and this turns out to be an empirically better alternative. Finally, we use policy gradient (Williams, 1992; Sutton et al., 1999) to maximize a weighted combination of the coherence and cohesion rewards. For illustrative purposes, we equally weigh them for updating our policy, i.e., the generator $G$.

| | |
|---|---|
| **source sentences** | the hotel inglaterra delivered as promised . the staff was welcoming and spoke good english . the cleaning staff did a very good job every day . the rooms were spotless and very modern . the bathroom was large and had a very nice shower , and there were two generously sized bath towels that were twice the size of normal towels . |
| $G_{\text{MLE}}$ | the breakfast in the morning was delicious and very good . it was the only hotel where i slept very well . the staff was very helpful in late afternoon or late times . the breakfast was adequate , with a decent range of cereals , fruit , and fruits . there is also free use of the coffee in the reception area . |
| $G_{\text{MLE+RL(coherence, cohesion)}}$ | the breakfast was plentiful including fresh breads and cooked to order . the location was fantastic . it is in the north of the marina and in a very short distance . the marina has a small swimming pool with sitting area and a small gym . they are very popular and guests have an evening reception which is very nice . |

Table 1: Sample generations from our MLE-trained baseline model, $G_{\text{MLE}}$, and our discriminator-guided model $G_{\text{MLE+RL(coherence, cohesion)}}$. The red texts highlight a common problem in $G_{\text{MLE}}$ - it exhibits a repetition, and an inconsistent opinion as a review. In contrast, our discriminator-guided model is able to generate a more interesting, and sentiment-consistent continuation.

## 5 EXPERIMENTS

In this section, we show results of training both $D_{\text{coherence}}$ and $D_{\text{cohesion}}$, and compare our RL-tuned generators $G_{\text{MLE+RL(cohesion)}}$, $G_{\text{MLE+RL(coherence)}}$, and $G_{\text{MLE+RL(coherence, cohesion)}}$ with the baseline model $G_{\text{MLE}}$. We argue that through the use of feedback from our simple discriminators to $G_{\text{MLE}}$, we improve the quality of generated texts. See Table 1 for a sample comparison.

### 5.1 DATASET AND EVALUATION METRICS

We use the publicly available TripAdvisor's hotel reviews dataset collected by Wang et al. (2010) and the Yelp review dataset[3]. We consider only subsets of the two review datasets satisfying the following two conditions: a review must have (1) at least 10 sentences, and (2) each sentence should have more than 5 and less than 30 words. This yields roughly 60,000 TripAdvisor reviews and 220,000 Yelp reviews, split into $[0.8, 0.1, 0.1]$ ratio for train/dev/test. We merge the source and target vocabularies, and limit it to the top 50,000 frequent words, excluding special tokens. For each of these reviews, as in Holtzman et al. (2018), we consider the first five sentences as the source input $S$ to $G$, and the following five sentences as the target output $T$ from $G$.

It is widely known that there is no accurate metric to evaluate the generator. Nevertheless, we report scores of standard metrics, such as negative log-likelihood (NLL), perplexity (PPL), BLEU and proportion of unique $n$-grams within a single generation (intra-unique-$n$), and across generations (inter-unique-$n$), as in Gu et al. (2018).

### 5.2 IMPLEMENTATION DETAILS

$G$ takes individual words as inputs and embeds into a pre-trained 300-dimensional word vectors from GloVe (Pennington et al., 2014). This embedding layer is fixed throughout training. $G$ uses a gated recurrent unit with two layers and a hidden size of 1024 for both bidirectional encoder and attention-based decoder. During optimization using Adam (Kingma & Ba, 2014), we set the learning rate to 2e-4 and clip the gradient's L2-norm to 1.0. We initially train $G_{\text{MLE}}$ by maximizing the word-level likelihood estimation (MLE) from data that consist of positive samples for 60 epochs on the TripAdvisor data and 30 epochs on the Yelp dataset, separately. These are our baseline models against which to empirically prove value of our hierarchical discriminators.

$D_{\text{coherence}}$ also uses the pre-trained GloVe word vectors[4], which are fixed. The source processing network and the target processing network have the same structure, but different parameters. If the encoder type is a CNN, the convolutional layer has filters of sizes 2, 3, 4, and 5, each with 512 filters. Each convolution filter is followed by a $\tanh$ activation. Then we max-pool in time over the features and append a fully connected layer into a feature embedding of dimension 512, followed by a batch normalization layer and a $\tanh$ activation. If the encoder type is a RNN, we use a 1-layered bi-directional GRU, concatenate final hidden states from both ends, and append the same network layers as used in the CNN-encoded counterpart. We use an Adam optimizer with a learning rate of

---

[3]https://www.yelp.com/dataset

[4]The vector dimension can be different from that of $G$. The differences were marginal for sizes 50, 100, and 300. For results shown in this paper, we used the same dimension of size 300.

| TripAdvisor | | Target Sentences Retrieval | | |
|---|---|---|---|---|
| Discriminators | Encoding | R@1 | R@5 | R@10 |
| $D_{\text{coherence}}$ | $\text{Conv}_{2,3,4,5}^{512}$ | 0.18 | 0.43 | 0.60 |
| | $\text{GRU}_{\text{1-layer, bi-dir.}}^{1024}$ | **0.26** | **0.50** | **0.65** |
| $D_{\text{cohesion}}$ | $\text{Conv}_{3,4,5,6}^{512}$ | **0.12** | **0.28** | **0.43** |
| | $\text{GRU}_{\text{1-layer, bi-dir.}}^{1024}$ | 0.11 | 0.21 | 0.33 |

| Yelp | | Target Sentences Retrieval | | |
|---|---|---|---|---|
| Discriminators | Encoding | R@1 | R@5 | R@10 |
| $D_{\text{coherence}}$ | $\text{Conv}_{2,3,4,5}^{512}$ | 0.33 | 0.61 | 0.74 |
| | $\text{GRU}_{\text{1-layer, bi-dir.}}^{1024}$ | **0.39** | **0.68** | **0.81** |
| $D_{\text{cohesion}}$ | $\text{Conv}_{3,4,5,6}^{512}$ | **0.14** | **0.33** | **0.47** |
| | $\text{GRU}_{\text{1-layer, bi-dir.}}^{1024}$ | 0.11 | 0.26 | 0.39 |

Table 2: Retrieval ratios for coherence and cohesion discriminators from a collection of 100 negative candidates. The reported numbers are averages over 20 evaluations. Notations: $\text{Conv}_{2,3,4,5}^{512}$ is a convolutional input encoder with filter sizes 2, 3, 4, and 5, and there are 512 filters for each filter size. $\text{GRU}_{\text{1-layer, bi-dir.}}^{1024}$ is a 1-layered bi-directional GRU input encoder with hidden size 1024. We experimented different configurations for both encoder types, and selected the best performing configurations.

1e-5. $D_{\text{cohesion}}$ is the same as $D_{\text{coherence}}$, except the CNN-encoded $D_{\text{cohesion}}$ has convolutional filters of sizes 3, 4, 5, and 6. We train both discriminators for 50 epochs and choose models with the best R@1 validation scores. Retrieval results are shown in Table 2.

In the tuning stage, we use the negative-critical sequence training as explained in Section 4 up to 5 epochs, with a learning rate of 1e-5. We also continue with supervised learning to $G$ to limit the policy search within a grammatically correct space, similar to Paulus et al. (2017); Wu et al. (2016); Lewis et al. (2017). In practice, sequence-level rewards are only available upon a completed generation, so they are sparse signals for the generator. Typically, sparse end-of-sequence rewards entail a noisy training, yet would want the learning generalize to the testing data. We observed that, for our particular task, most noises were caused by exploration, and the learning generalized to the testing data. Thus, reward shaping was unnecessary, unlike previous works (Li et al., 2017; Yang et al., 2018) that further provided signals for partially generated sequences. For all generations, we used greedy decoding since we did not see a significant difference using beam search. Results are shown in Table 3.

| | Model | NLL | PPL | BLEU-3 | BLEU-4 | BLEU-5 | intra-unique-1 | intra-unique-2 | inter-unique-2 | inter-unique-3 | length ratio |
|---|---|---|---|---|---|---|---|---|---|---|---|
| **TripAdvisor** | $G_{\text{MLE}}$ (baseline) | 0.86 | 2.36 | 0.38 | 0.19 | 0.08 | 0.66 | 0.93 | 0.40 | 0.72 | 1.08 |
| | $G_{\text{MLE +RL(cohesion)}}$ | **0.77** | **2.18** | **0.46** | **0.27** | **0.14** | 0.64 | 0.94 | 0.38 | 0.71 | 0.97 |
| | $G_{\text{MLE+RL(coherence)}}$ | 0.80 | 2.24 | 0.44 | 0.25 | 0.12 | 0.64 | 0.94 | 0.39 | 0.72 | 1.06 |
| | $G_{\text{MLE+RL(coherence, cohesion)}}$ | 0.80 | 2.25 | 0.44 | 0.24 | 0.12 | 0.65 | 0.94 | 0.40 | 0.72 | 1.02 |

| | Model | NLL | PPL | BLEU-3 | BLEU-4 | BLEU-5 | intra-unique-1 | intra-unique-2 | inter-unique-2 | inter-unique-3 | length ratio |
|---|---|---|---|---|---|---|---|---|---|---|---|
| **Yelp** | $G_{\text{MLE}}$ (baseline) | 1.32 | 3.84 | 0.37 | 0.17 | 0.07 | 0.68 | 0.95 | 0.54 | 0.86 | 1.07 |
| | $G_{\text{MLE+RL(cohesion)}}$ | 1.26 | 3.65 | **0.45** | **0.23** | **0.11** | 0.68 | 0.95 | 0.53 | 0.85 | 1.05 |
| | $G_{\text{MLE+RL(coherence)}}$ | **1.24** | **3.56** | 0.45 | 0.23 | 0.11 | 0.69 | 0.95 | 0.55 | 0.87 | 1.00 |
| | $G_{\text{MLE+RL(coherence, cohesion)}}$ | 1.25 | 3.59 | 0.43 | 0.22 | 0.11 | 0.69 | 0.95 | 0.56 | 0.88 | 1.05 |

Table 3: An ablation study with automated evaluation metric scores: NLL, PPL, BLEU-$n$, intra/inter-unique-$n$, along with the length ratio with the length of corresponding true target sentences as 1. Results show that our proposed discriminators helped improve notably in BLEU scores, NLL and PPL, with marginal difference in diversity. We used equally weighted rewards, and significant numbers are highlighted in **bold** before rounding.

## 5.3 DISCUSSION

In Table 2, notice that for RNNs outperform CNNs for coherence models, and CNNs outperform RNNs for cohesion models. One explanation is that RNNs are effective in encoding a sequential input yet exhibit drawbacks when encoding into hidden states at both ends of a *long* input, otherwise well-known as a long-range dependency problem.

To aid understanding of the roles of $D_{\text{coherence}}$ and $D_{\text{cohesion}}$, we show randomly selected positive and negative samples and corresponding rewards in Table 4.

While reinforcing coherence and cohesion properties in text generation through surrogate models is an important research direction, we consider our results to be preliminary, and our experiment results allude to room for improvement, such as recall scores. This is because our methods to

| source | cohesion | coherence |
|---|---|---|
| this hotel was unbelievably overpriced . | 0.0002 | |
| we were looking for something cheaper but thought we would at least be staying in a decent hotel having paid that much when booking . | 0.0411 | |
| it wasn t clear when booking that we would have to share a bathroom . | 0.0084 | |
| there was one shower for the whole floor which was tiny and unclean . | 0.0054 | |
| the room was old and lacking in facilities . | | |
| **target** | | |
| the beds were very uncomfortable and the linen was very old . | 0.0768 | |
| breakfast was ok , but the staff were incompetent . | 0.0591 | |
| on our last day they were too lazy to clean our table and never bothered taking our order . | -0.0097 | **+0.3735** |
| we had to leave having had no breakfast , as we ran out of time . | 0.0457 | |
| they saw us get up and leave and didn t even apologise for the appalling lack of service . | | |
| **negative target** | | |
| the staff recommended great restaurants with very reasonable prices within walking distance . | 0.0514 | |
| the paris hop on bus stops nearby . | 0.0798 | |
| the gare l est is within 3 blocks . | -0.0156 | |
| we paid 75 euro per nite excluding breakfast but paid for breakfast one day and found it very good and reasonably priced . | 0.0082 | **-0.2001** |
| the rooms are clean and bathrooms ensuite . | | |
| **more examples of cohesion** | | |
| once you get there you are greeted by the staff . they explain everything to you , and in english , not the best , but good enough . | 0.1004 | |
| the coffee was even good for a coffee snob like myself . the hotel is much smaller than i thought and only has six floors . | -0.1103 | |
| the only negative was the curtain in the bathroom . it was very shear and we felt that people in the building across the street could look right in at night . | 0.0787 | |
| the beer at the lobby bar was stale . there are many friendly cats on the grounds . | -0.0830 | |

Table 4: Coherence and cohesion margin scores on test data. The cohesion score at the end of each line is computed with its next sentence. This is an example of contradiction and inconsistent sentiment, suggestive of incoherence. We append more examples with extreme cohesion margin scores.

construct negative samples from an unlabelled dataset are not thorough. For example, a randomly mismatched sentence that follows a given sentence may actually be a valid continuation. In this work, we overlook this problem since our proposed schemes are shown to be effective in modeling coherence and cohesion.

# 6 CONCLUSION

In this paper, we propose to model coherence and cohesion through simple training mechanisms via models parametrized by neural networks, and quantify coherence and cohesion into negative-critical margin scores. The coherence discriminator $D_{\text{coherence}}$ provides a macro-level view on structuring a multi-sentence text. It assesses how likely two text chunks form a coherent paragraph, using sentence-level features. On the other hand, the cohesion discriminator $D_{\text{cohesion}}$ provides a micro-level view on local connectivity between sentence pairs. It assesses how cohesive two consecutive sentences are, using word-level features.

The margin scores computed by these discriminators are used as reward signals for training neural language models via policy gradient. Empirical results on two long-form text generation tasks show that our surrogate coherence and cohesion models, trained through simple yet effective methods, help improve over the strong baseline, an attention-based bidirectional MLE-trained sequence-to-sequence model in a number of automatic metrics.

Future work will focus on casting the long-form text generation task using the GANs framework. In this framework, the coherence and cohesion discriminators are modified against model-generated texts, and in turn, provide signals to learn neural language models.

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
