# OpenReview forum: "A bird's eye view on coherence, and a worm's eye view on cohesion"
_ICLR.cc/2019/Conference_

### Official Review · AnonReviewer3 · 2018-11-01
**Interesting proposal to use discriminators to model coherence in NLG, but completely ignores prior work and presentation is confusing**

**Rating:** 4
**Confidence:** 4

**Review:**

The idea of training discriminators to determine coherence and cohesion, and training those discriminators as part of an NLG system using policy gradients, is an interesting one. However, there are two major problems with the papers as it stands:

1) it completely ignores the decades of NLG literature on this topic before the "neural revolution" in NLP;
2) the presentation of the paper is confusing, in a number of respects (some details below).

To claim that this is the first paper to capture cross-sentence linguistic properties for text generation is the sort of comment that is likely to make experienced NLG researchers very grumpy. A good place to start looking at the extensive literature on this topic is the following paper:

Modeling Local Coherence: An Entity-Based Approach, Barzilay and Lapata (2007)

One aspect in which the presentation is muddled is the order of the results tables. Table 2 is far too early in the paper. I had no idea at that point why the retrieval results were being presented (or what the numbers meant). You also have cohesion in the table before the cohesion section in 3.2. Likewise, Table 1, which is on p.2 and gives examples of system output, is far too early.

Perhaps the biggest confusion for me was the difference between cohesion and coherence, and in particular how they are modeled. The intro does a good job of describing the two concepts, and making the contrast between local and global coherence, but when I was reading 3.1 I kept thinking this was describing cohesion ("T that follows S in the data" - sounds local, no?). And then 3.2 seems to suggest that coherence and cohesion essentially are being modeled in the same way, except shuffling happens on the word level? I suppose what I was expecting was some attempt at a global model for coherence which goes beyond just looking at consecutive sentence pairs.

I wonder why you didn't try a sequence model of sentences (eg bidirectional LSTM). These are so standard now it seems odd not to have them.

Do you describe the decoding procedure (greedy? beam?) at test time anywhere?

I liked Table 4 and found the example pairs with the scores to be useful qualitative analysis.

"Based on automated NLP metrics, we showed a significant improvement" - which metrics? not clear to me that the improvements in Table 3 are significant.

Minor presentation points
--

"followed by a logically sound sentence" - might want to rephrase this, since you don't mean logical soundness in a technical sense here (I don't think).

The comment in the conclusion about being "convinced" the architecture generalizes well to unseen texts is irrelevant without some evidence.

---

> ### Author Response · Authors · 2018-11-26
> **Author response to AnonReviewer3**
>
> Thank you for your thoughtful comments. Your comments have been helpful.
>
> We agree that our tables and figures are presented in a confusing manner. We revised our submission to make our presentation clear.
>
> (a)
> “Perhaps the biggest confusion for me was the difference between cohesion and coherence, and in particular how they are modeled. The intro does a good job of describing the two concepts, and making the contrast between local and global coherence, but when I was reading 3.1 I kept thinking this was describing cohesion ("T that follows S in the data" - sounds local, no?). And then 3.2 seems to suggest that coherence and cohesion essentially are being modeled in the same way, except shuffling happens on the word level? I suppose what I was expecting was some attempt at a global model for coherence which goes beyond just looking at consecutive sentence pairs.“
>
> We would like to clarify how our coherence and cohesion models work. The input to the coherence model is a pair of sequence of `sentence’ embeddings (from source text chunk and target text chunk), whereas the input to the cohesion model is a pair of sequence of `word’ embeddings (from two consecutive sentences). This is the fundamental difference between the coherence and cohesion models. Thus the coherence model has a global view to judge an entire paragraph. On the other hand, the cohesion model has a local view to judge any two neighboring sentences.
>
> In section 3.1, “T that follows S in the data” contains the global coherence relation, because T and S are multi-sentence text chunks. In section 3.2, consecutive sentence pair (s_{i,1} and s_{i,2}) contains the local cohesion relation.
>
> (b)
> “I wonder why you didn't try a sequence model of sentences (eg bidirectional LSTM). These are so standard now it seems odd not to have them. “
>
> We chose the CNN encoder because this is the standard architecture in relevant text generation works using GANs, such as SeqGAN, TextGAN or LeakGAN. However, we took this opportunity to test bidirectional RNN and posted new results.
>
> Notice that for RNNs outperform CNNs for coherence models, and CNNs outperform RNNs for cohesion models. One explanation is that RNNs are effective in encoding a sequential input yet exhibit drawbacks when encoding into hidden states at both ends of a `long’ input, otherwise well-known as a long-range dependency problem.
>
> (c)
> “Do you describe the decoding procedure (greedy? beam?) at test time anywhere?”
>
> We used greedy decoding at test time. When we experimented with a larger beam size, we did not see significant difference in text generation quality and it was more computationally intensive.

---

### Official Review · AnonReviewer1 · 2018-11-03
**overall weak evaluation and too many unsubstantiated claims**

**Rating:** 2
**Confidence:** 4

**Review:**

The paper proposes a method for improving the quality of text generation by optimizing for coherence and cohesion. The authors develop two discriminators--a "coherence discriminator" which takes as input all of the sentence embeddings (i.e. averaged word embeddings) of the document and assigns a score, and a "cohesion discriminator" which takes as input the word embeddings of two consecutive sentences and assigns a score. In the former, the score is the cosine similarity between the encodings of the first and second half of the document. In the latter, the score is the cosine similarity between the encodings of the two sentences. Both discriminators use CNNs to encode the inputs. The discriminators are trained to rank true text over randomly drawn negative samples, which consist of randomly permuted sentence orderings and/or random combinations of first/second half of documents. This discriminators are then used to train a text generation model. The output of the text generation model is scored by various automatic metrics, including NLL, PPL, BLEU, and number of unique ngrams in the outputs. The improvements over a generically-trained generation model are very small.

Overall, I did not find this paper to be convincing. The initial motivation is good--we need to find a way to capture richer linguistic properties of text and to encourage NLG to produce such properties. However, the discriminators presented do not actually capture the nuances that they purport to capture. As I understand it, these models are just being trained to incentivize high cosine similarity between the words in the first/second half of a document (or sentence/following sentence). That is not reflective of the definitions of coherence and cohesion, which should reflect deeper discourse and even syntactic structure. Rather, these are just models which capture topical similarity, and naively at that. Moreover, training this model to discriminate real text from randomly perturbed text seems problematic since 1) randomly shuffled text should be trivially easy to distinguish from real text in terms of topical similarity and 2) these negative samples are not (I don't think) at all reflective of the types of texts that the discriminators actually need to discriminate, i.e. automatically generated texts. Thus, even ignoring the fact that I disagree with the authors on exactly what the discriminators are/should be doing, it is still not clear to me that the discriminators are well trained to do the thing the authors want them to do. I have various other concerns about the claims, the approach, and the evaluation. A list of more specific questions/comments for the authors is below.

- There are a *lot* of unsubstantiated claims and speculation about the linguistic properties that these discriminators capture, and no motivation of analysis as to how they are capturing it. Claims like the following definitely need to be removed: "learn to inspect the higher-level role of T, such as but not limited to, whether it supports the intent of S, transitions smoothly against S, or avoids redundancy", "such as grammar of each of the sentences and the logical flow between arbitrary two consecutive sentences"
- You only use automated metrics, despite acknowledging that there is no good way to evaluate generation. Why not use human eval? This is not difficult to carry out, and when you are arguing about such subtle properties of language, human eval is essential. There is no reason that BLEU, for example, would be sensitive to coherence or cohesion, so why would this be a good way to evaluate a model aimed to capture exactly those things?
- Also related to human eval, there should be an intrinsic evaluation of the discriminators. Do they correlate with human judgments of coherence and cohesion? You cannot take it for granted that they capture these things (I very much believe they do not), so present some evidence that the models do what you claim they do.
- The reported improvements are minuscule, to the extent that I would read them as "no difference". The only metric where there is a real difference is on number of unique ngrams generated cross inputs, which is presumably because its just learning (being encouraged to) spit out words that were in the input. I'd like to see the baseline of just copying the input as the output.
- You mention several times that these models will pick up on redundancy. It is not clear to me how they could do that. Aren't they simply using a cosine similarity between feature vectors? Perhaps I am missing something, but I don't see how this could learn to disincentivize redundancy but simultaneously encourage topical similarity. Could you explain this claim?

---

> ### Author Response · Authors · 2018-11-26
> **Author response to AnonReviewer1 (1)**
>
> Thank you for your comments.
>
> We would like to clarify our methodology. Apparently, our AnonReviewer2 also had the same misunderstanding so we append the same example to further help you understand our approach. We revised our writing to make this point clear.  We would like to respond to your comments hereafter.
>
> (a)
> “As I understand it, these models are just being trained to incentivize high cosine similarity between the words in the first/second half of a document (or sentence/following sentence). That is not reflective of the definitions of coherence and cohesion, which should reflect deeper discourse and even syntactic structure. Rather, these are just models which capture topical similarity, and naively at that. “
>
> On one hand, our model has the potential to capture the coherence (cohesion) correlation between two halves of a paragraph. If the source and the target networks shared the parameters, then your intuition is indeed correct.  Quoting your words, the discriminators would be “trained to incentivize high cosine similarity between the words in the first/second half of a document” and simply “capture topical similarity, and naively at that”.  However, coherent and cohesive paragraphs have particular “syntactic structures” which we are glad that you mentioned, therefore we modeled feature extraction through convolutional layers and process the first and second half of a paragraph with different networks (yet same architecture).  Differences in parameter weights in the convolution layer and fully connected layers will govern how semantic and syntactic features are extracted for either half of the paragraph.
>
> Regarding the use of cosine similarity, each source and target in a sample pair is processed through different networks which have the same architecture but different parameters. See Figure 1 for the illustration. Therefore, the example of two pairs provided in AnonReviewer2's comments in fact give two ‘different’ cosine similarity scores.  This technique, in various forms, has been applied in information retrieval, web search ranking, image captioning to list a few. See Huang et al. CIKM 2013.
>
> Here we provide an example taken from the TripAdvisor dataset and scored through our pre-trained cohesion model.
>
> it is full of homeless people . however , they did not bother us .  →  0.14
>
> however , they did not bother us . it is full of homeless people .  →  -0.12
>
> On the other hand, it is possible that the correlation we capture is not coherence (or cohesion) which people typically have in mind, because our approach is learning purely from raw data. We showed examples from our trained coherence (or cohesion) model, and showed that its rating indeed aligns well with our typical impression; see our experiment section. Therefore, our coherence (or cohesion) models indeed learn some correlation that align well with the coherence (or cohesion) concept.
>
> I hope this answers your concerns.
>
> (b)
> “Moreover, training this model to discriminate real text from randomly perturbed text seems problematic since 1) randomly shuffled text should be trivially easy to distinguish from real text in terms of topical similarity … You mention several times that these models will pick up on redundancy. It is not clear to me how they could do that. Aren't they simply using a cosine similarity between feature vectors? Perhaps I am missing something, but I don't see how this could learn to disincentivize redundancy but simultaneously encourage topical similarity. Could you explain this claim?  “
>
> Let us consider generating three types of negative samples for coherence discriminator. Given a batch of source text and target text pairs, we mismatch the source and target pairs, which is what we mean by ‘rotating target texts with source texts fixed’. Also given the true source text and target text pair, we shuffle the sentence-wise orders of the target text. Finally, combine for previous two methods.
>
> Although we admit that some negative samples may be trivially easy to distinguish, some other negative samples may be difficult if only the sentence order is perturbed, even difficult to us humans.

---

> > ### Author Response · Authors · 2018-11-26
> > **Author response to AnonReviewer1 (2)**
> >
> > (c)
> > “these negative samples are not (I don't think) at all reflective of the types of texts that the discriminators actually need to discriminate, i.e. automatically generated texts.”
> >
> > We admit that the name "discriminator" is confusing. We use the discriminator in a RL fashion, instead of a GAN fashion. In the RL framework, these negative samples to train the ``discriminator’’ do not need to reflect the automatically generated texts. Our loss is MLE loss + RL loss. The RL reward only plays a role of regularizer. More precisely, during the training, the discriminator rewards the generated sentences similar to real data and penalizes generated sentences that are similar to our constructed negative examples. The MLE loss part penalizes all samples, including the automatically generated texts, that are not the same with the training data.
> >
> > It is straightforward to extend our RL framework to a GAN framework, where the discriminators are jointly trained with the generator, as we pointed out in our future work. In this case, the discriminator will directly use the generated sentences as negative examples and the reward function even does not need the MLE part any more.
> >
> > (d)
> > “You only use automated metrics, despite acknowledging that there is no good way to evaluate generation. Why not use human eval? This is not difficult to carry out, and when you are arguing about such subtle properties of language, human eval is essential. There is no reason that BLEU, for example, would be sensitive to coherence or cohesion, so why would this be a good way to evaluate a model aimed to capture exactly those things?”
> >
> > Yes, we agree that human evaluation is essential and we are currently working on this. To answer your concern, in fact, our coherence and cohesion models do have implications on BLEU as they are trained to provide rewarding or penalizing signals. Notice that cohesion model in particular sees constructed negative samples with lower BLEU scores, thus able to provide rewarding or penalizing signals. The coherence model assesses how sentences are organized and the generator model, which is continually trained via MLE, is regularized modifying its policy and mimicking data distribution.
> >
> > (e)
> > “Thus, even ignoring the fact that I disagree with the authors on exactly what the discriminators are/should be doing, it is still not clear to me that the discriminators are well trained to do the thing the authors want them to do. … Do they correlate with human judgments of coherence and cohesion?“
> >
> > It is possible that the correlation we capture is not coherence (or cohesion) which people typically have in mind, because our approach is learning purely from raw data. We showed examples from our trained coherence (or cohesion) model, and showed that its rating indeed aligns well with our typical impression; see our experiment section. Therefore, our coherence (or cohesion) models indeed learn some correlation that align well with the coherence (or cohesion) concept.

---

### Official Review · AnonReviewer2 · 2018-11-03
**Missing relevant comparisons, evaluations, and references**

**Rating:** 2
**Confidence:** 4

**Review:**

This paper addresses long-text generation, with a specific task of being given a prefix of a review and needing to add the next five sentences coherently.  The paper proposes adding two discriminators, one trained to maximize a cosine similarity between source sentences and target sentences (D_{coherence}) and one trained to maximize a cosine similarity between two consecutive sentences.  On some automatic metrics like BLEU and perplexity, an MLE model with these discriminators performs a little bit better than without.

This paper does not include any manual evaluation, which is critical for evaluating the quality of generated output, especially for evaluating coherence and cohesion.  This paper uses the task setup and dataset from "Learning to Write with Cooperative Discriminators", Holtzman et al., ACL 2018.  That paper also includes many specified aspects to improve the coherence (from the abstract of that paper "Human evaluation demonstrates that text generated by our model is preferred over that of baselines by a large margin, significantly enhancing the overall coherence, style, and information of the generations.").  But this paper:
--Does not compare against the method described in Holtzman et al., or any other prior work
--Does not include any human evaluations, even though they were the main measure of evaluation in prior work.

This paper states that "To the best of our knowledge, this paper is the first attempt to explicitly capture cross-sentence linguistic properties, i.e., coherence and cohesion, for long text generation."  There is much past work in the NLP community on these.  For example, see:
 "Modeling local coherence: An entity-based approach" by Barzilay and Lapata, 2005 (which has 500+ citations).
It has been widely studied in the area of summarization, for example,
"Using Cohesion and Coherence Models for Text Summarization", Mani et al., AAAI 1998, and follow-up work.
And in more recent work, the "Learning to Write" paper that the dataset and task follow from addresses several linguistically informed cross-sentence issues like repetition and entailment.

The cosine similarity metric in the model is not very well suited to the tasks of coherence and cohesion, as it is symmetric, while natural language isn't.  The pair:
"John went to the store to buy some milk."
"When he got there, they were all out."

and

"When he got there, they were all out."
"John went to the store to buy some milk."

would have identical scores according to a cosine similarity metric, while the first ordering is much more coherent than the second.

The conclusion says "we showed a significant improvement": how was significance determined here?

---

> ### Author Response · Authors · 2018-11-26
> **Author response to AnonReviewer2**
>
> Thank you for the comments and mentioning relevant prior work.
>
> (a)
> We believe there are some misunderstanding about  the main technique that underlies our proposed discriminators. Regarding the use of cosine similarity, each source and target in a sample pair is processed through different networks which have the same architecture but different parameters. See Figure 1 for the illustration. Therefore, the example of two pairs provided in your comments in fact give two ‘different’ cosine similarity scores.  This technique, in various forms, has been applied in information retrieval, web search ranking, image captioning to list a few. See Huang et al. CIKM 2013.
>
> Here we provide an example taken from the TripAdvisor dataset and scored through our pre-trained cohesion model.
>
> it is full of homeless people . however , they did not bother us .  →  0.14
>
> however , they did not bother us . it is full of homeless people .  →  -0.12
>
> We revised our writing appropriately.
>
> (b)
> Our model is not directly comparable to Learning2Write (L2W) model for a number of reasons. There are differences in the architecture. For example, we do not use adaptive softmax for modeling the probability distribution of the vocabulary, nor do we have the same vocabulary implementation structures and word embedding functions. Yet, these architectural differences can be solved via engineering. The more fundamental problem is that we cannot use the modified beam search decoding scheme as proposed in their work. In L2W, the modified beam search decoding scheme is integrated with their discriminators, and is coded for only a simple sample input, and no batch sample which disables creating negative sample batch for our discriminators at decoding time. Also our discriminators provide scores on fully generated sequences rather than partially generated sequence. These key differences in scoring and sampling process make it difficult to compare. They do not provide the full data so we are currently working on our own implementation to compare with, and show efficacy of our discriminators.
>
> (c)
> Thank you for letting us know the missed prior work. We have modified our writing accordingly. We are currently working on human evaluation.

---

### Author Response · Authors · 2018-11-26
**Addressing the common misunderstanding**

Let us elaborate, through examples, on the types of negative samples the discriminators are trained to discriminate. Consider the two real reviews, A and B, taken from our dataset.

Review A (1/2): we had several recommendations from friends where to stay , but opted to take a chance here . the reviews sounded quite positive and convincing . we were not disappointed . the room was small but very clean , and we were able to store our luggage . the shower was hot with great water pressure , but it was a handheld shower which makes a quick shower difficult .

Review A (2/2): the location could not be beat . we were able to walk so many places and the metro access was right nearby . the location was quiet because it was tucked in just off the beaten path . the staff was the best part . they were able to help us find our way anywhere and made great recommendations for dinner .

Review B (1/2): after reading various reviews , i decided to stay at the best western downtown while visiting my daughter in vancouver . after one night , i knew it was the wrong place for me . the room was tiny although clean with nice amenities . the noise from the street was terrible in spite of the fact that i had a sleep machine running . i like a dark room to sleep in , and light streamed through the edges of the curtains .

Review B (2/2): worst of all , the neighborhood made me uncomfortable , not dangerous but panhandlers and homeless people on the street corners . i have never been uncomfortable walking in downtown vancouver , but i did in this area . i was also very disappointed that there was no complimentary breakfast , only an overpriced , average restaurant attached to the hotel . considering all the good places to eat in vancouver , you don t want to waste your money at the white spot restaurant . we checked out as soon as possible and moved to the hampton inn , which is where we should have stayed from the beginning .

Suppose you read [Review B (1/2), Review A(2/2)] and [Review A (1/2), Review B (2/2)]. These are aspects of incoherence that the coherence discriminator learns to pick up by seeing a large number of positive samples and such negative samples. In fact this is one example of incoherence. To address your concern, some other negative samples exhibit redundancy and the discriminator learns to distinguish through modifying source and target networks that have different parameters.

---

### Meta-Review · Area_Chair1 · 2018-12-14

**Confidence:** 4
**Recommendation:** Reject

**Metareview:**

This paper attempts at modeling coherence of generated text, and proposes two kinds of discriminators that tries to measure whether a piece of text is coherent or not.

However, the paper misses several related critical references, and also lacks extensive evaluation (especially manual evaluation).

There is consensus between the reviewers that this paper needs more work before it is accepted to a conference such as ICLR.